# Potential Cell-Based and Cell-Free Therapy for Patients with COVID-19

**DOI:** 10.3390/cells11152319

**Published:** 2022-07-27

**Authors:** Marselina Irasonia Tan, Nayla Majeda Alfarafisa, Popi Septiani, Anggraini Barlian, Mochamad Firmansyah, Ahmad Faizal, Lili Melani, Husna Nugrahapraja

**Affiliations:** 1School of Life Sciences and Technology, Institut Teknologi Bandung, Bandung 40132, Indonesia; popi.septiani@itb.ac.id (P.S.); aang@sith.itb.ac.id (A.B.); firman@sith.itb.ac.id (M.F.); afaizal@itb.ac.id (A.F.); lili@sith.itb.ac.id (L.M.); husna_np@itb.ac.id (H.N.); 2Department of Biomedical Sciences, Faculty of Medicine, Universitas Padjadjaran, Sumedang 45363, Indonesia; nayla.alfarafisa@unpad.ac.id

**Keywords:** mesenchymal stem cells (MSCs), cell-based therapy, cell-free therapy, SARS-CoV-2, COVID-19, secretome, exosome

## Abstract

Since it was first reported, the novel coronavirus disease 2019 (COVID-19) remains an unresolved puzzle for biomedical researchers in different fields. Various treatments, drugs, and interventions were explored as treatments for COVID. Nevertheless, there are no standard and effective therapeutic measures. Meanwhile, mesenchymal stem cell (MSC) therapy offers a new approach with minimal side effects. MSCs and MSC-based products possess several biological properties that potentially alleviate COVID-19 symptoms. Generally, there are three classifications of stem cell therapy: cell-based therapy, tissue engineering, and cell-free therapy. This review discusses the MSC-based and cell-free therapies for patients with COVID-19, their potential mechanisms of action, and clinical trials related to these therapies. Cell-based therapies involve the direct use and injection of MSCs into the target tissue or organ. On the other hand, cell-free therapy uses secreted products from cells as the primary material. Cell-free therapy materials can comprise cell secretomes and extracellular vesicles. Each therapeutic approach possesses different benefits and various risks. A better understanding of MSC-based and cell-free therapies is essential for supporting the development of safe and effective COVID-19 therapy.

## 1. Introduction

Since the discovery of mesenchymal stem cells (MSCs) in 1968 by Friedenstein et al. (1970), stem cell-based therapy showed enormous potential in treating various diseases [1,2]. In the early study, Friedenstein et al. (1970) described MSCs as an adherent fibroblast-like population, likely originated from the mesoderm primary germ line. Friedenstein and colleagues isolated MSCs from bone marrow and found that MSCs could differentiate into bone cells. In addition, they observed that MSCs could secrete hematopoietic growth factors and cytokines [3]. Later, researchers found that MSCs could be isolated from other adult tissues, such as adipose and peripheral blood, and extraembryonic tissues, such as umbilical cord and placental. Furthermore, an appropriate progenitor can induce MSCs to differentiate into multiple cell types, such as osteoblasts, adipocytes, and chondroblasts [2]. Therefore, MSCs are an excellent treatment option for various clinical pathologies. To date, there was much research on MSC-based products since the first paracrine effect of MSCs was reported by Haynesworth et al. (1996) [4]. MSC-derived paracrine was shown to promote several biological functions on injured tissue, such as angiogenesis, neovascularization, and arteriogenesis, eventually protecting the organ from an acute injury [5]. In addition, MSC-based products, widely known as cell-free therapy, have fewer immunogenicity effects and a longer shelf life than cell-based therapies [6]. MSC-based therapy was used to treat viral infections, such as HIV-1 [7] and influenza [8]. Therefore, it may be effective against the newly emerged virus.

The novel coronavirus disease 2019 (COVID-19) was first reported in Wuhan, China, at the end of 2019. Later, the World Health Organization (WHO) named the virus-associated disease SARS-CoV-2 and declared the disease a global public health emergency [9]. Various therapeutic options for COVID, including chemical agents, such as remdesivir and tocilizumab, convalescent plasma therapy, and repurposed drugs, were explored [10]. Nevertheless, physicians and intensivists still have limited treatment options for treating patients with COVID-19. Despite the available vaccines and drugs, many patients still need to rely on their immune system. Moreover, supportive treatment is only given for complications [11]. In severe cases, patients can experience a profound inflammatory and multi-organ dysfunction known as the cytokine storm syndrome (CSS) that causes acute respiratory distress syndrome (ARDS) and acute lung injury, resulting in multiple organ failure, extensive pulmonary inflammation, edema, fibrosis, and even death [12,13].

There are three types of stem cell therapeutic approaches. First, cell-based therapy involves the direct use and injection of stem cells into the target tissue or organ; sometimes, stem cells also differentiate into desired cells. Second, tissue engineering uses stem cells and biodegradable scaffolds. Stem cells are cultured and differentiated into the desired tissue before implantation into the target tissue or organ. Third, cell-free therapy uses the secreted products of cells to stimulate local tissue self-regeneration, communication, and refinement [14].

An increasing interest in the therapeutic benefits of MSCs for patients with COVID-19 led to multiple clinical trial investigations. In addition, MSC-free therapy may be a potent treatment for treating patients with COVID-19 and preventing complications. As of 31 October 2021, there are 366 ongoing clinical trials worldwide on using stem cells to treat COVID-19 (https://clinicaltrials.gov/ct2/home, accessed on 31 October 2021). The present review summarizes the current understanding of the possible interactions between MSCs and SARS-CoV-2 in cell-based and cell-free therapeutic applications. MSC-based therapy directly uses MSCs, either autologous or allogenic stem cells, by transplanting or infusing MSCs into patients with COVID-19. On the other hand, MSC-free therapy encompasses the use of MSC secretomes, MSC extracellular vesicles, and MSC miRNAs. We also discuss the possible advantages and disadvantages of each therapy, while including the clinical evidence of the therapies, which may strengthen the hypothesis of MSC utility as a new therapeutic option for patients with COVID-19.

## 2. SARS-CoV-2

The novel coronavirus disease 2019 (COVID-19) is caused by severe acute respiratory syndrome coronavirus 2 (SARS-CoV-2) [15] of the *Coronaviridae* family, a group of viruses characterized by an enveloped phospholipid bilayer and crown-like spike proteins throughout the membrane [16]. There are four subfamilies in *Coronaviridae*, α, β, γ, and δ. Only the α and β subfamilies can infect humans. Like SARS and MERS, two previously discovered major coronaviruses, SARS-CoV-2 belongs to the β sub-family [17]. Morphologically, coronaviruses are spherical or pleomorphic membrane virions with a diameter of 50–200 nm [16]. In addition, SARS-CoV-2 consists of four structural proteins, spike (S), membrane (M), envelope (E), and nucleocapsid (N), with diameters of 60–140 nm [17]. Tyrrell and Bynoe (1966) found that *Coronaviridae* had a positive single-stranded RNA with a genomic of 26–32 kb [18]. The virus’ genomic sequence comprises 14 open reading frames (ORFs) that encode 16 nonstructural proteins, 9 accessory proteins, and 4 structural proteins [19]. SARS-CoV-2 morphology can be seen in Figure 1 below. The origin of the virus remains controversial. Nevertheless, several reports suggest that SARS-CoV-2 is 96% whole-genome identical to a bat coronavirus. Therefore, it is most likely that this virus comes from a bat, a known natural reservoir of coronavirus [20].

Compared with other coronaviruses, SARS-CoV-2 is more contagious, spreading at a remarkably rapid rate through air droplets or fecal media [21], while surviving for up to 3 h in the air [9]. The SARS-CoV-2 incubation period can be relatively long, ranging between 2 and 14 days [22]. For replication and shedding, SARS-CoV-2 invades the host and hijacks its cellular machinery [16]. Although the mechanisms underlying the pathogenesis of SARS-CoV-2 remain unknown, patients with COVID-19 display pathological features, such as hypoxemia (low blood oxygen), chronic pulmonary inflammation, edema, and diffuse alveolar damage [23]. However, most of the infected patients are asymptomatic or have mild symptoms. Only around 15% of the patients experience pneumonia and ARDS, and only 5% of them progress to multiple organ dysfunctions [24]. The mortality rate varies between 5% and 18%, depending on the preventive measures taken [25].

The first targets of SARS-CoV-2 for viral entry are the cells in the respiratory tract, airway epithelial cells, alveolar epithelial cells, vascular endothelial cells, and alveolar macrophages of host tissues [19]. The spike protein, which covers the viral membrane, plays a significant role in SARS-CoV-2 transmission. The spike protein has a receptor-binding domain (RBD) that mediates direct contact with angiotensin-converting enzyme 2 (ACE2), a host cell-surface receptor. Its S1/S2 polybasic cleavage site is then cleaved by cathepsin L and transmembrane protease serine 2 (TMPRSS2). Cathepsin L and TMPRSS2 also mediate virus–cell membrane fusion and endosomal compartment trafficking, delivering viral particles into the cell [19]. After SARS-CoV-2 enters the host cell cytosol, the virus will start the translation of their ORFs (ORF1a and ORF 1b) into the replicase that eventually forms the RNA-dependent RNA-polymerase [26]. The replicase components will rearrange the host’s endoplasmic reticulum into double-membrane vesicles (DMVs) to facilitate the replication of viral genomic and subgenomic RNAs (sgRNA). The positive RNA strands serve as a template for negative-strand RNA and sgRNA. All structural and accessory proteins are translated from the sgRNAs and packed in the ER–Golgi intermediate compartment (ERGIC) with a positive-sense RNA genome to form new virions [19]. The viral life cycle is also portrayed in Figure 1 below.

Currently, various therapeutic options, approved by the FDA under Emergency Use Authorization (EUA) for treating patients with COVID-19, are available. For example, patients can be treated with antiviral drugs, anti-SARS-CoV-2 monoclonal antibodies, anti-inflammatory drugs, or immunomodulatory agents [27]. Treatments are specifically chosen based on the severity of symptoms and possible risk factors. Nevertheless, these treatments are ineffective in preventing complications. In addition, the effectiveness of a treatment varies among patients.

SARS-CoV-2 is equipped with crown-like spike proteins throughout the membrane, which play a significant role in SARS-CoV-2 transmission. Viral entry is facilitated by S protein which has a receptor-binding domain (RBD) that directly interacts with the host cellular receptors angiotensin-converting enzyme 2 (ACE2) and cell transmembrane serine protease TMPRSS2, followed by viral uptake and membrane fusion. The virus starts the translation of their two large open reading frames (ORF1a and ORF1b) into the replicase that eventually forms the RNA-dependent RNA-polymerase. The replicase components will rearrange the host’s endoplasmic reticulum into double-membrane vesicles (DMVs) to facilitate the replication of viral genomic and subgenomic RNAs (sgRNA). All translated structural and accessory protein are translocated into the ER–Golgi intermediate compartment (ERGIC) and start to generate new virions. Finally, virions are secreted by the exocytosis mechanism.

## 3. Mesenchymal Stem Cells for Cell-Based Therapy

Research related to MSCs began in the late 1960s after they were discovered by Soviet scientist Alexander Friedenstein, who demonstrated that clonogenic progenitor cells in mouse bone marrow could give rise to many mesodermal cell lineages. In addition, the clonogenic progenitor does not belong to the hematopoietic cell lineage. According to Friedenstein, MSCs likely originate from the mesoderm primary germ layer; therefore, they can generate into various mesenchymal cell lineages [28,29]. The term “MSCs” was first coined by Arnold Caplan, who associated these cells with another stem cell from mesodermal tissues in the embryo. However, it took almost a decade before researchers could finally describe the original identity, tissue distribution, frequency, and natural function of MSCs [28]. Shortly after discovering a method for isolating and expanding MSCs, the field grew rapidly. Several studies found that MSCs could be isolated from embryonic or adult tissues. For example, MSCs were identified in the cranial neural crest of embryonic tissues [30]. In addition, we can isolate MSCs from bone marrow, umbilical cord blood (UCB) [31], placental tissue [32], adipose tissue [33], and dental pulp in adults [34].

The increasing interest in using MSCs for research and clinical study raises the need for a broadly accepted definition of MSCs. The International Society for Cellular Therapy (ISCT) recommends ‘multipotent mesenchymal stromal cell’ as the standard definition. In addition, ISCT defines four minimum criteria for MSCs for research purposes [35], as seen in Table 1 below.

The distinctive characteristics of MSCs make them an ideal source of therapeutic cells for repairing connective tissue trauma. Researchers hypothesized that the MSC main role in repairing injured tissue was migrating into the wounded location, engrafting, differentiating into the needed functional cell, and regenerating a new part of the tissue. However, intensive research showed that MSCs repaired injured tissues mainly by modifying the microenvironment in an injured tissue through paracrine activities, cell–cell interactions, and extracellular vesicle secretions [29]. MSCs also possess several advantageous properties. For example, they are relatively easy to isolate and expand in standard culture conditions and can be stored for a long period without losing their potency. More importantly, MSC treatment does not cause any adverse effects; therefore, it is considered safe for clinical therapeutic applications [36]. To date, various applications of MSCs for clinical purposes were studied with promising results.

According to Saeedi et al. (2019), MSCs have several possible modes of action during their therapeutic applications. Firstly, during migration, MSCs possess a homing capacity, or a special ability to migrate toward the site of injury through chemoattraction. The chemotactic gradient resulting from inflammation facilitates the movement of MSCs. On the site of injury, selectin and integrin facilitate the adhesion of MSCs. Eventually, MSCs infiltrate into the inflammation site and become involved in regeneration. Secondly, MSCs have a role in tissue repair and regeneration. Besides their homing capacity, MSCs can also differentiate into specialized cells. In addition, MSCs secrete several paracrine signals to modify the microenvironmental conditions in the injured site. Thirdly, MSCs exert immunomodulatory and anti-inflammatory effects. They suppress inflammation by regulating B-cells, T-cells, and dendritic and natural killer cells. They also regulate neutrophil infiltration and cytokine secretion while upregulating the expression of the genes responsible for phagocytosis, bacterial clearance, and the downregulation of inflammatory cytokines. Overall, MSCs can modulate inflammatory activity by controlling the level of inflammatory cytokines on the site of injury. Fourthly, MSCs have an anti-apoptotic activity. They inhibit apoptotic activity by upregulating DNA repair mechanisms, downregulating mitochondrial death signaling activity, balancing the oxidative level of antioxidant and oxidant molecules, and regulating pro-apoptotic gene expression. Fifthly, MCSs can promote neo-vascular growth. They promote gene expression of angiogenic cytokines to improve endothelial cell growth and vascular regeneration. Sixthly, MCSs can activate resident stem cells. They secrete many chemoattractant molecules, such as cell-derived factor-1α (SDF-1α), to target endogenous MSCs or recruit endogenous immune cells to initiate tissue repair. Lastly, MCSs have antimicrobial effects. They can secrete peptides with antimicrobial properties on the site of injury to disrupt the bacterial membrane, eventually suppressing bacterial activity [37].

Although cell-based therapy is the most common MSC therapeutic application, it presents several problems. According to Nguyen et al. (2016), it remains difficult to trace the fate and survival of MSCs in the recipient, increasing the risk of spontaneous differentiation into undesirable cells [38]. There is also a need to develop another method of specifically isolating and characterizing MSCs since they are highly heterogeneous and difficult to characterize. In addition, the optimal protocol for ex vivo expansion, mode of delivery, optimum dosage, and administration frequency remains unclear [39]. Moreover, genetic aberrations and manipulations during ex vivo expansion can compromise normal cell function and cause complications. Furthermore, MSCs have a tendency to tumor growth, as stem cells and cancer cells respond to similar types of growth factors; this can potentially create an environment for tumorigenesis [40].

Despite the problems mentioned above, much effort was conducted to overcome these problems. Yin et al. (2019) revealed critical factors for maintaining MSC properties [41]. The following factors should be examined when growing MSCs in vitro for cell therapy: (1) MSC isolation and purification procedure from various sources; (2) culture media selection; (3) culture appliance or system control; (4) the standardized quality assays for in vitro senescence and genetic stability; and (5) the appropriate disease-specific mode of action and potency assays. Several studies also showed that optimized culture conditions can convert heterogeneous MSCs into relatively homogeneous cells of identical size, phenotype, and differentiation potential [42]. Lian et al. (2016) developed protocols for generating high-quality MSC from iPSC, which expressed specific MSC surface markers, had adult bone marrow-derived MSC (BM-MSCs) properties, and had a high proliferative capacity. Moreover, with this protocol, MSCs were successfully up-scaled for tissue engineering and cellular therapeutics development [43]. Bloor et al. (2020) produced iPSC-derived MSCs using an optimized, manufacturing process compliant with good manufacturing practice (GMP) and used these cells in phase I clinical trial for patients with steroid-resistant acute graft versus host disease (SR-aGvHD) [44]. Nevertheless, to improve MSC quality for cell therapy, a standard characterization assay is needed to determine the identity, purity, sterility, and potency of MSCs used to obtain optimal cells [45].

## 4. Mesenchymal Stem Cells for Cell-Free Therapy

Although MSCs show promising therapeutic potential for various ailments, several studies suggest that MSC therapeutic benefits are not limited to cell regeneration. MSCs vanish after several days post-transplantation, resulting in low effectivity [46]; this finding suggests that the efficacy of MSC therapy does not rely on the physical proximity of the injected cells [47]. Several studies confirm that MSC therapeutic efficacy is derived from the synergistic action of the biomolecules it secretes [48]. Still, the clinical application of MSCs through transplantation (MSC-based therapy) faces several obstacles, causing the renewed interest in MSC-free therapy. Secretomes, extracellular vesicles (especially exosomes), and miRNAs are commonly used in the cell-free application of MSCs.

### 4.1. Secretome Therapy

Secretomes are diverse factors or biomolecules secreted by cells into the extracellular space [49]. Secretomes are enriched with various bioactive molecules, such as lipids, proteins, nucleic acid, chemokines, cytokines, growth factors, hormones, extracellular vesicles that could exert paracrine effects upon neighboring cells to carry out cells biological function, including cellular migration, proliferation, immunomodulation, and tissue regeneration [50]. Nevertheless, the composition of secretomes varies among individual cells and tissues since physiological states and pathological conditions, such as hypoxic cultures, added chemical stimuli, and 3D cultured conditions (spheroids) [46], can exhibit significant effects [49].

According to Ferreira et al. (2018), MSC mechanisms are greatly influenced by diverse extrinsic factors [46]. Nevertheless, only a few studies examined the effect of MSC preconditioning on the MSC secretome profile. Several papers conclude that hypoxia can enhance the regenerative and cytoprotective effects of MSC-CM [51,52,53,54,55]. Meanwhile, Ma et al. (2009) suggest that hypoxic culture is relevant to the in vivo physiological condition of MSCs [56]. Therefore, MSC cultures display a positive response to low-oxygen tension preconditioning. On the other hand, preconditioning with inflammatory cytokines promotes the production of immunomodulatory biomolecules that help MSCs modulate inflammation [57]. In addition, the 3D culturing method (or spheroids) can increase the production of cell survival biomolecules compared with the conventional culture methods [58]. Again, this culture creates a microenvironment that resembles the in vivo physiological condition of MSCs since spheroids create a hypoxic condition for the inner layers of the MSC culture [59].

MSC secretome therapy without optimization may qualify MSC therapeutic efficacy. Therefore, knowing the best MSC culture condition or treatment is crucial for obtaining the most appropriate secretome. Several investigations reported that the MSC culture conditions affect the cytokines in the secretome produced by the MSC. Some cytokines and growth factors contained in the secretome of MSC are necessary for therapeutic purposes. Cytokines such as TGFβ-1, VEGF, FGF, HGF, and PDGF contained in secretomes from hADSC or KGF, HGF, PDGF, and stromal cell-derived factor-1 (SDF-1) from MSCs are helpful for wound healing. However, some cytokines and factors secreted from MSCs, such as tumor necrosis factor-α (TNF-α) and interleukin-6 (IL-6), are unsuitable for therapy. Thus, optimizing the MSC culture is crucial because some cytokines and factors are secreted by the MSCs. Repressed Rap1 expression in MSC cell culture can downregulate cytokines, such as tumor necrosis factor-α (TNF-α) and interleukin-6 (IL-6) production [60,61].

As mentioned above, the hypoxia condition enhance regenerative and cytoprotective effects of MSC-CM. However, hypoxia can accumulate ROS in cells, causing aging-associated cellular dysfunction. Liang et al. (2018) showed that that in hypoxia, MSC aging was related to the downregulation of ERBB4 expression. Conversely, overexpressing ERBB4 in aging MSCs could increase telomere length and telomerase activity, further rejuvenating MSC cells [62]. Another modification that can be performed is overexpression of HO-1 in MSC to enhance therapeutic efficiency [63]. HO-1, a cytoprotective enzyme, plays an essential role in the body’s defense system’s anti-oxidative, anti-inflammatory, and anti-apoptotic effects. Hypoxia also downregulates the expression of HO-1 [64].

A therapeutic application using MSC secretomes has advantages over the application using transplanted MSCs [49]. Firstly, unlike the MSC-based therapy, secretomes do affect the protein synthesis/productions level, thus decreasing the likelihood of immunogenicity. Secondly, MSC secretomes significantly suppress the need for cell number, thus avoiding the potential alteration caused by MSC expansion and storage. Thirdly, a high-output, efficient production of MSC secretomes is possible to achieve. Fourthly, an application using MSC secretomes is more profitable from an economic perspective. Fifthly, MSC secretome therapy can be easily tailored to exert specific properties and effects. Sixthly, the safety and efficacy of MSC secretomes are relatively easy to evaluate. Lastly, MSC secretomes are relatively easy to store without losing much potency or increasing toxicity from cryopreservative agents. Therefore, MSC secretomes emerge as a promising cell-free therapeutic tool for COVID-19 due to the positive results from the studies with MSC-conditioned media for treating acute and chronic lung diseases. MSC secretomes also possess the same anti-inflammatory, immunomodulatory, regenerative, pro-angiogenic, and anti-protease properties as the parental MSCs. In addition, MSC secretomes potentially act on several cytokines synergistically and simultaneously [65,66].

### 4.2. Exosome Therapy

Extracellular vesicles (EVs) are lipid bilayer membrane vesicles produced by apoptotic cells, or released from cells through multivesicular bodies or cell membrane shedding [67]. All cells can secrete various types of EVs, including MSCs. EVs can be found in almost every type of biological fluid, such as plasma, serum, saliva, amniotic fluid, breast milk, and urine [68]. One key role of EVs is facilitating cell-to-cell communication, an essential hallmark for multicellular organisms [68]. EVs can transfer proteins, RNAs, DNAs, carbohydrates, and lipids as messenger biomolecules for a targeted cell to elicit pleiotropic responses [67]. Various mechanisms, such as plasma membrane fusion [69], endoplasmic reticulum scan and sorting into lysosome [70], endosome membrane fusion [71], and endosomal rupture [68] can facilitate the release of cargo from EVs into a targeted cell.

The ISEV consensus recommends using “extracellular vesicle” as the generic term for the naturally released particles from the cell demarcated by the lipid bilayer membrane. This particle cannot multiply by itself since it does not contain any functional organelle [72]. Nevertheless, researchers usually classify EVs into different types based on size, biogenesis, and functions, namely exosomes, microvesicles, apoptotic bodies, ectosomes, and oncosomes, to prevent confusion within a more specialized field [68]. EV classification can be seen in Figure 2 below.

Extracellular vesicles are commonly classified according to their size and biogenesis. Exosomes are EVs with a size range of 30–150 nm, microvesicle (MV) diameters range from 50 to 1000 nm, while apoptotic bodies have a diameter of 1000–5000 nm. Exosomes are derived from the endosomal pathway; they are developed from the repeated invagination of multivesicular bodies (MVBs). On the other hand, MVs are generated from the sites with high membrane blebbing and involve contractile apparatus to initiate vesicular pinching. Apoptotic bodies result from the fragmentation of the cells undergoing apoptosis.

#### 4.2.1. Exosomes

Exosomes are EVs with a size range of 30–150 nm playing a critical role in cell–cell communication [73]. According to another study, exosomes have a diameter less than 100 nm with a density of 1.10–1.18 g/mL [74]. According to several studies (e.g., [75]), exosomal protein content can be specific or non-specific to certain cell types. On the other hand, exosomal lipid content is usually conserved and restricted [76]. Exosomes are derived from the endosomal pathway; they are developed from the repeated invagination of multivesicular bodies (MVBs). At the early stage, endosomes fuse with endocytic vesicles to recycle, degrade, or exocytose their content. Endosomes will undergo a sequence of alterations until they become late endosomes, or multivesicular bodies (MVBs). During maturation, the cargo encapsulated within MVBs is sorted. Eventually, MVBs fuse with lysosomes or the plasma membrane to be released as exosomes [74].

#### 4.2.2. Microvesicles

Derived from the plasma membrane, microvesicles (MVs) are biological vesicles with a diameter of 50–300 nm [74] or 100–1000 nm [77]. Nevertheless, MVs are characterized by the presence of phosphatidylserine in the outer layer of the membrane [74]. MV density ranges from 1.04 to 1.07 g/mL. MVs contain biomolecules, such as plasma membrane proteins, cytosolic proteins, nucleic acids, and many other metabolites [78]. MVs are involved in extracellular alteration, intracellular signaling, and cell invasion through cell-independent matrix proteolysis [74]. Slightly different from exosomes, MVs are generated from sites with high membrane blebbing. Since MVs originate by plasma pinching, they are exposed, from time to time, to cytoplasmic materials. MV biogenesis starts with the vertical trafficking of molecular cargo to the plasma membrane, followed by the redistribution of membrane lipid and the recruitment of contractile apparatus to initiate vesicular pinching. Lastly, MV formation is stimulated in cells invading through appropriate matrix conditions [78,79].

#### 4.2.3. Ectosomes

According to Riazifar et al. (2017), microvesicles and ectosomes are the same molecules [78]. Nevertheless, Gurunathan et al. (2021) specify ectosomes as biological vesicles with a diameter of 50–500 nm [68]. Like exosomes, ectosomes play a crucial role in intracellular communication; yet ectosome secretion occurs by the accumulation of cargo at the cytosolic surface near the plasma membrane. In contrast, exosome secretion occurs through the budding of the outer cell membrane (pinching), similar to MVs. Ectosome secretion relies on local microdomain assembly and the induction of GTPase ARF6 and small GTPase for cortical actin activation [68].

#### 4.2.4. Apoptotic bodies

Apoptotic bodies (Abs) were first described by Kerr in 1974 as vesicle-like structures resulting from cell fragmentation during apoptosis. ABs have a diameter of 1000–5000 nm and a density between 1.16 and 1.28 g/mL. ABs are the main feature of apoptotic cells since they result from the fragmentation of the cells undergoing apoptosis. Apoptotic bodies are composed of partially hydrolyzed cellular components. These EVs play a key role in cellular homeostasis as they are able to remove apoptotic cells and immunomodulation [68,74,78].

Exosomes are the most prominent among the various types of EVs [5]. MSC-derived exosomes were first isolated in 2010. Isolated exosomes were used to repair myocardial ischemic/reperfusion in mice [80]. Due to their unique properties, MSC-derived exosomes serve as potential biomarkers and effective therapy for many diseases and ailments [74]. Compared with MSC-based therapy, MSC-derived exosomes are safer and have a longer shelf life [81]. MSC-derived exosomes are also relatively easy to collect and target tissues or microenvironments specifically [5]. MSC-derived exosomes are enriched with important bioactive molecules that regulate the phenotype, function, survival, and homing of immune cells, including mRNA and miRNA molecules, enzymes, cytokines, chemokines, and growth factors [82]. Their lipid bilayer membrane is also composed of cholesterol, sphingomyelin, ceramide, and lipid raft proteins [83] that facilitate trafficking and membrane fusion, regardless of biological barriers [82].

Meanwhile, exosome secretion is controlled by several factors. Hypoxia and the overexpression of the tetraspanin CD9 and hypoxia-inducible-factor-1α genes increase exosome production [73]. As mentioned previously, MSC-derived exosomes possess several advantages over MSC-based therapy. From an economic point of view, using MSC-derived exosomes enables the development of cheaper and more effective treatments than the conventionally manufactured cell-based therapies [84,85]; this point is certainly worthy of consideration in terms of developing therapies during a global pandemic, such as COVID-19.

## 5. MSC-Based Therapy for SARS-CoV-2

The broad and multifactorial mode of action of MSC therapeutic applications is ideal for overcoming various pathological symptoms of COVID-19. Previous studies produced promising results for MSC therapies in tackling ARDS [86] and sepsis [8,87], common critical pathological symptoms in patients with COVID-19. The main symptom of ARDS is the disruption of alveolar endothelial and epithelial cells, which leads to inflammation and decreasing pulmonary permeability. Furthermore, patients will experience edema, intrapulmonary shunting, and hypoxemia. MSCs can prevent the condition from worsening by modulating inflammatory activity and preventing apoptosis [88]. According to Johnson et al. (2016), human MSC treatment can improve ARDS and sepsis in mice and rats [89]. While there is no conclusive evidence from human clinical trials to date, the phase 1 trial by Wilson et al. [90] and Zheng et al. [91] showed that administration of MSCs did not cause any adverse events. The aforementioned mode of action by MSCs, such as immunomodulatory effects and anti-inflammatory activity, also bolsters the capacity of MSCs as a potential candidate treatment for SARS-CoV-2 infection since these features can promote the protection of alveolar epithelial cells during viral infection [9]. In addition, MSCs have a low level of major histocompatibility complexes (MHCs), while they do not synthesize ACE2 and TMPRSS2 receptors. A low level of MHCs prevents MSCs causing a patient immunogenic reaction [88].

Several pilot studies using MSC therapy to overcome the pathological symptoms of patients with COVID-19 produced promising results. Leng et al. found that transplanting MSCs at 1 × 10^6^ cells/kg body weight could improve the clinical outcomes of patients with severe or mild COVID-19 symptoms; MSCs exerted an immunomodulatory effect on the patient 2 days after transplantation [92]. In addition, MSC treatment was not associated with adverse effects. Guo et al. also reported that therapy with MSCs isolated from the umbilical cord could improve the clinical outcome of COVID-19 patients with severe symptoms by improving patient oxygenation levels and suppressing cytokine storm incidents [93]. Liang et al. (2020) discovered that the allogenic treatment of the MSCs isolated from the umbilical cord could depress inflammatory symptoms in patients with COVID-19 pneumonia [94]. Using MSCs from the umbilical cord in their clinical trials, Feng et al. [95] and Lanzoni et al. [96] also obtained positive results from patients with COVID-19. Feng et al. observed better oxygenation indexes, cytokine levels, radiological images, and lymphocyte counts, and a significant decline in mortality rate, in COVID-19 patients 28 days after MSC transplantation [95]. On the other hand, Lanzoni et al. observed a significant decrease in mortality rate and recovery time after an IV injection with 100 × 10^6^ of MSCs in patients with COVID-19. Both studies also confirmed the safety of MSC therapy. With an increasing number of clinical trials producing positive results, MSCs are a strong candidate for a broader clinical application to accelerate the recovery of patients with COVID-19 [96].

In general, MSCs regulate severe to mild COVID-19 symptoms (specifically ARDS and sepsis) through three main mechanisms of action: (1) immunomodulatory effects, (2) reparative or recovery effects, and (3) antimicrobial effects. MSCs modulate the inflammatory environment by directly secreting anti-inflammatory soluble factors such as IL-6, IL-10 [97,98], transforming growth factor β (TGF-β) [97,98], and prostaglandin E2 (PGE2) [98]. As shown in previous studies using lipopolysaccharide-induced acute lung injury and cecal ligation and puncture (CLP)-induced sepsis mouse models, MSCs rebalanced the inflammatory condition by using these capacities [97,98], thereby reducing the occurrence of cytokine storms [95,96]. The modulatory effect of MSCs during sepsis and ARDS also involves regulations of innate and adaptive immune cell responses [87], such as monocyte/macrophages [99], dendritic cells [100], neutrophils [101], and regulatory T cell populations [102]. Numerous studies also shown that MSCs are able to protect pulmonary endothelial and epithelial cells from damage induced by inflammation. MSCs secrete several cell-protective bioagents that played an important role in cellular protection and restoration, such as Ang1, PGE2, HGF, KGF, and VEGF [88,103,104]. As an example, in vitro studies showed that MSCs restored the epithelial protein permeability after the induction of cytomix injury by secreting paracrine soluble factor Ang1. Cytomix is a mixture of human IL-1β, TNF-α, and IFN-γ that is frequently utilized as a surrogate for ALI pulmonary edema fluid [105]. Ang1 mediated the cytoskeletal re-organization of actin and claudin-18 in epithelial cells and restored epithelial permeability [106]. Moreover, MSCs possessed anti-apoptotic and anti-oxidant activity which also bolster the reparative effect of MSCs [87]. An in vivo study showed that MSCs transplantation could attenuate LPS-induced and burn-induced organ injury by reducing cell death through the activation of the Akt1 signaling pathway [107]. Lastly, several studies also showed that MSCs could improve bacterial clearance in ARDS and sepsis models [87,108]. In his study, Gupta et al. (2012) found that MSCs could enhance bacterial clearance by upregulating the expression of lipocalin 2 in pneumonic mice models. Lipocalin 2 is an antibacterial protein with an important role in the host defense mechanism of bacterial infection [108]. Other antibacterial protein molecules, such as beta defensin-2 (BD2) [109] and Syndecan-2 positive [110], were also overexpressed and mediated the MSC bacterial clearance capacity. On the other hand, in vivo studies using gram-negative sepsis mice conclude that MSC action on bacterial clearance is mediated through the improvement of mononuclear cell phagocytic activity [111] and through the upregulation of heme oxygenase-1 expression in macrophages [112].

## 6. MSC-Free Therapy for SARS-CoV-2

Currently, there are nine ongoing studies on MSCs specifically using MSC secretomes or MSC-conditioned media to overcome COVID-19 symptoms (as of 3 August 2021, https://clinicaltrials.gov/ct2/home). According to Bari et al., MSC-derived secretomes tend to show similar therapeutic results and efficacy to MSC therapeutic applications [65]. It is also suggested that the dosage of MSC secretomes should be adjusted, taking into account the continuous release of secretomes by whole-cell MSCs in clinical trials. Nevertheless, secretomes have several notable therapeutic effects that make them a valid option for accelerating the recovery of patients with COVID-19. Khan et al. revealed that bleomycin-injured rats treated with MSC secretomes had an improved lung architecture, and suppressed α-SMA and collagen content, concluding that MSC secretomes were enriched with many bioactive substances with anti-inflammatory, immunosuppressive, and angiogenic properties that helped the lung to recover [113]. Felix et al. also demonstrated that rats with bleomycin-induced lung injury demonstrated better regeneration after treatment with MSCs and MSC-conditioned media [114]. Furthermore, both MSCs and MSC-conditioned media could modulate in situ immune responses and prevent the development of pulmonary fibrosis. Pati et al. showed that an MSC-CM could restore the permeability of pulmonary endothelia in in vitro and in vivo analyses. In addition, MSC-conditioned media also inhibit the systemic levels of inflammatory cytokines and chemokines in the serum of treated rats [115]. Furthermore, Goolaerts et al. found that an MSC-CM could reverse epithelial hyperpermeability and epithelial Na+ transport, preventing the development of acute lung injury. Therefore, with their advantageous properties, MSC secretomes represent a suitable approach for accelerating the recovery of COVID-19 patients with severe symptoms [116].

Various studies also clearly demonstrated the positive effect of MSC-derived exosomes on COVID-19-related pathological symptoms. The intratracheal and intravenous administration of MSC-derived EVs can reduce lung inflammation and edema in bacteria-induced acute lung injury (ALI) mice [117]. ALI and ARDS, major complications in COVID-19 patients with severe symptoms, can lead to acute hypoxemic respiratory failure [21,84]. Monsel et al. used *E. coli*-induced pneumonia mice to test the effect of the administration of MSC-derived EVs. They found that MSC-derived EVs could suppress inflammatory cytokine secretion, increase lung protein permeability, and improve the survival rate of the mice through the KGF-mediated effect [118]. Another study also showed that MSC-derived exosomes could enhance the phagocytosis of macrophages and the secretion of TNF-α and IL-8, ameliorating LPS-induced lung injuries in mice [119]. In addition, the intratracheal administration of MSC-derived EVs can improve lung inflammation and restore alveolar-capillary permeability in mice previously induced by LPS to stimulate ALI condition [120]. Further to this, Khatri et al. reported that MSC-derived EVs had anti-influenza activity by inhibiting influenza virus replication and virus-induced apoptosis in lung epithelial cells. Moreover, influenza virus infection was significantly reduced in the lung of influenza-infected pigs after the intratracheal administration of MSC-EVs [121]. In addition, researchers also considered the potential interplay of micro-RNA (miRNA), an important biomolecule found in the extracellular vesicles of MSCs. Several studies reported that miRNAs have a special host–virus interaction during viral infection. miRNAs and viruses develop special cross-talk so that either can alter the host’s transcriptome or indirectly regulate viral infections, resulting in pro or antiviral effects [122]. Host miRNAs can bind onto the 3′-UTR [123,124] or 5′UTR of the infected SARS-CoV-2 virus, suggesting an important interplay between the host and the virus. Park et al. (2021) identified at least five miRNAs (miR-92a-3p, miR-26a-5p, miR-23a-3p, miR-103a-3p, and miR-181a-5p) isolated from mesenchymal stem-cell-derived EVs that could prevent cytopathic effect of SARS-CoV-2 virus and the suppression of pro-inflammatory responses in the infected cells [125].

Several mechanisms may contribute to the immunomodulatory and reparative effect of MSC cell-free therapy. First, soluble factors produced by MSCs could act as a mediator for the beneficial impact of MSCs on SARS-CoV-2-related symptoms. As shown by Pati et al. (2011), MSCs-derived factors could restore pulmonary endothelial cell permeability by preserving adherent junctions between endothelial cells [115]. Fang et al. (2010) stated that these actions were likely due to the secretion of angiopoietin-1. MSC-secreted paracrine soluble factor angiopoietin-1 eventually mediated the cytoskeletal re-organization of human alveolar epithelial cells and restored the permeability between cells [106]. On the other hand, Goolaerts et al. (2014) found that MSCs secreted keratinocyte growth factor (KGF) that modulates the protective effect of MSCs on alveolar epithelial sodium transport after hypoxic and cytomix-induced lung injury [116]. Second, MSCs could interact with target cells through EV paracrine mechanism. Secreted EVs contained various important biomolecules that induce regulatory responses and restore homeostasis in target cells or tissue [126]. MSCs-derived EVs are able to carry lipids, various kinds of RNA, enzymes, transcription factors, signals, receptors, and even organelles, such as mitochondria. Through the receptor-binding process, EVs interface with targeted cells followed by either membrane cell fusion or an endocytosis mechanism to deposit their content [127]. Several matrix remodeling enzymes carried by EVs, such as matrix metalloproteinase (MMPs), heparanases, hyaluronidase, and tissue inhibitors of metalloproteinases (TIMPs), were able to modulate the remodeling and tissue repair process of an alveolar injury [128,129]. Interestingly, EVs were also able to transfer MSC-derived mitochondria into the targeted cell and restore the cellular bioenergetics function. In fact, mitochondrial transfer mechanisms produce relatively long-term reparative effects [87]. Mitochondrial transfer from MSCs to T cells was proven to modulate the pro-inflammatory signature of T cells. This mechanism also induced T cell activation, differentiation, and function to regulated inflammatory response [130]. In addition, EV-derived MSCs also could transfer mRNA and miRNA for modulating their immunomodulatory and reparative effect. A study by Zhu et al. (2013) demonstrated that the protective effect of MSCs in injured alveolar epithelium was primarily mediated by the transfer of the KGF mRNA mechanism [117]. Tang et al. (2017) also suggested that the therapeutic effects of MSC derived-EVs in endotoxin-induced ALI were mediated partly by the transfer of Ang-1 mRNA content [120].

In their COVID-19 research study, Kuate et al. developed a novel vaccine approach by incorporating the S protein from SARS-CoV-2 into MSC-derived exosomes [131]. The administration of S-containing exosomes induced the development of neutralizing antibody titers. Meanwhile, Sengupta et al. studied the safety and efficacy of ExoFloTM, an MSC-derived exosome produced under good manufacturing standards [132]. They found that the intravenous administration of ExoFloTM could improve the clinical status and oxygenation of patients with COVID-19. ExoFloTM also exhibited an immunomodulatory effect and caused no adverse effects. Wang et al. (2022) developed an inhalable-vaccine candidate containing a recombinant SARS-CoV-2 RBD conjugated to lung-derived exosomes. The lung-derived exosomes enhanced the RBD uptake by APC, which can improve the effectivity of this vaccine candidate in mice. On the other hand, this vaccine candidate reduced severe pneumonia and inflammatory infiltrates in hamsters [133]. Popowski et al. (2022) also revealed that lung-derived extracellular vesicles (Lung-Exo) were superior cargo for mRNA and protein vaccines. Lung-Exo enhanced candidate vaccine distribution and retention in the respiratory tract [134]. Many ongoing clinical trials use MSC-derived exosomes instead of MSCs to combat SARS-CoV-2. Although MSC-derived exosomes require more extensive investigation, MSC-derived exosome therapy still warrants consideration as a potential solution for preventing deterioration in patients with COVID-19.

## 7. Conclusions

Since its first case was reported, the COVID-19 pandemic affected more than a hundred million people worldwide, spreading rapidly and affecting every aspect of human civilization. COVID-19 is caused by severe acute respiratory syndrome coronavirus 2 (SARS-CoV-2), which belongs to the β sub-family of *Coronaviridae*, a group of viruses with a phospholipid bilayer envelope and crown-like spike proteins throughout the membrane. At the time of writing, the search for effective and efficient COVID-19 therapies remains the most significant and complex task for all scientists. Various therapeutic options, including chemical agents, convalescent plasma therapy, and repurposed drugs, are being explored. Nevertheless, there are no effective treatments available. As a result, patients must rely on their immune system and supportive treatment is only given if complications occur.

MSC, or mesenchymal stromal cell, therapies are expected to fulfill the need for supportive treatments to alleviate severe COVID-19 symptoms, while accelerating patient recovery. There is much progress in the research and many studies produced promising results. MSCs, among the most widely used stem cells, exert various positive effects on human health. The distinctive characteristics of MSCs make them an ideal source of therapy for repairing connective tissue trauma. Currently, there are two main groups of stem cell therapeutic approaches, cell-based therapies that directly use and inject MSCs into target tissues or organs and cell-free therapies that use stem-cell-secreted products as the primary material. In the context of COVID-19 therapeutic practice, an MSC-based therapy directly uses MSCs, either autologous or allogenic, by transplanting or infusing them into patients with COVID-19. On the other hand, an MSC-free therapy can use MSC secretomes and MSC EVs to overcome a range of symptoms and complications of SARS-CoV-2 infection. The mechanisms by which MSCs exert their effects are diverse. For instance, MSC cell therapy regulates mild to severe COVID-19 symptoms through three main mechanisms of action: (1) immunomodulatory effects, (2) reparative or recovery effects, and (3) antimicrobial effects. For MSC cell-free therapy, however, the mechanisms of action mainly involve (1) the secretion of soluble factor, (2) transfer of mitochondria, and (3) transfer of mRNA and/or miRNA via extracellular vesicles into targeted cells or tissue. Each therapeutic approach has different benefits as well as risks, as briefly depicted in Figure 3.

Cell-based therapies and cell-free therapies have different benefits and risks that need careful consideration for the development of safe and effective COVID-19 therapies. In several instances, preclinical research supports both therapies individually with various beneficial effects. Unfortunately, at the time of writing, there is still no gold standard assay capable of precisely predicting MSC therapeutic efficacy. In addition, there is no clear conclusion regarding several MSC limitations, such as safety, tumorigenicity, pro-fibrogenicity, and heterogeneity. Clinical scale feasibility, time consumption, and production costs should also be considered. Careful consideration of the risks and benefits of each therapy is reasonable in this early-phase research. Standardizations are also necessary for optimizing the results and minimizing the heterogeneity in patient outcomes. A better understanding of MSC-based and cell-free therapy is critical for supporting the development of safe and effective COVID-19 therapies. The clinical use of MSC therapies to treat COVID-19 has a long way to go and needs much refinement. Nevertheless, there are promising reports and results to apply and consider, which brings hope to combating the pandemic and meeting the urgent medical needs.

## Figures and Tables

**Figure 1 cells-11-02319-f001:**
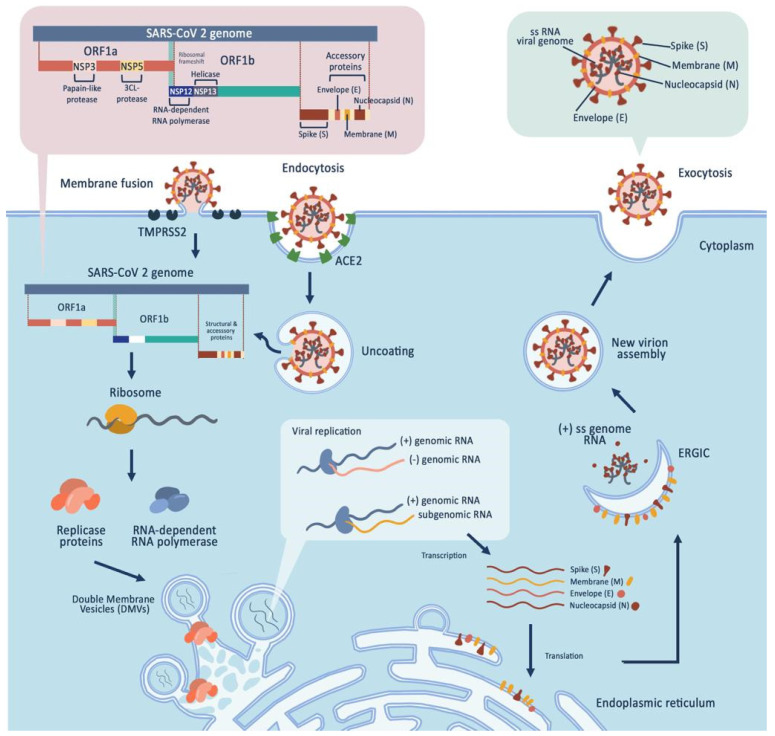
SARS-CoV-2 morphological features and life cycle.

**Figure 2 cells-11-02319-f002:**
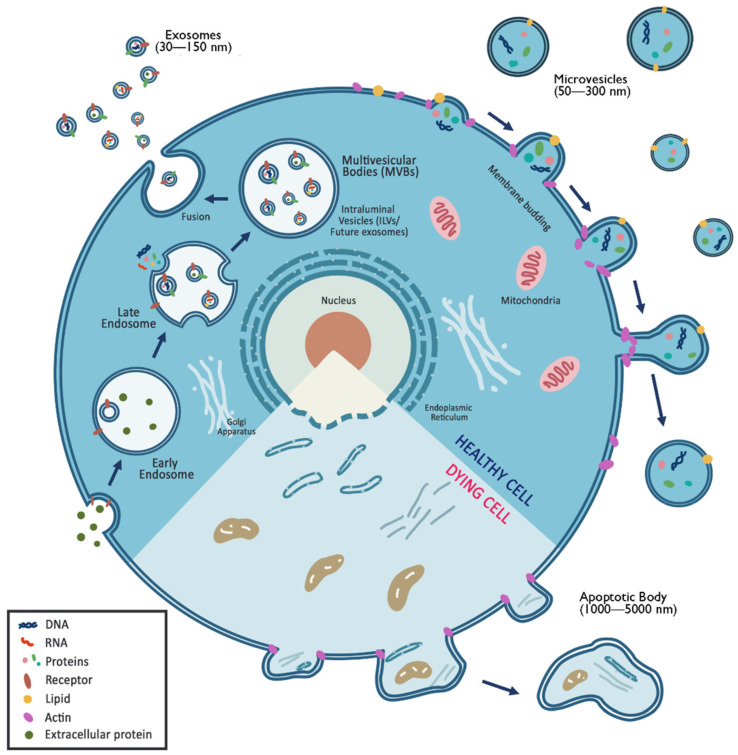
General classification and biogenesis of extracellular vesicles.

**Figure 3 cells-11-02319-f003:**
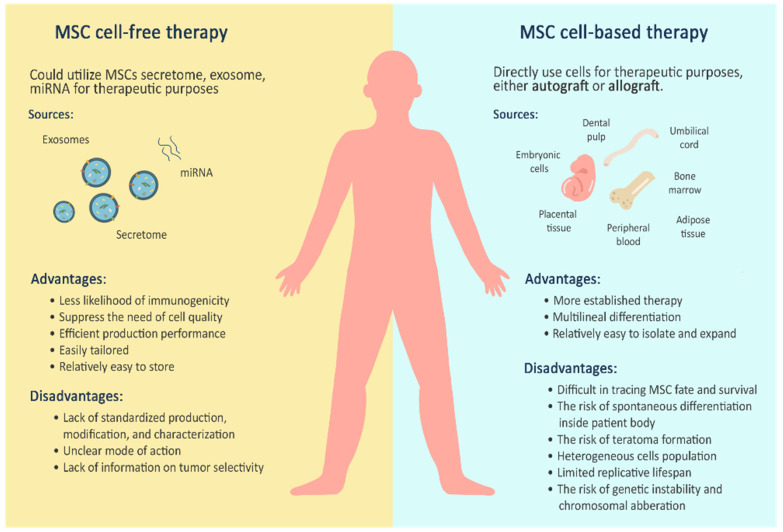
Two main groups of stem cell therapeutic approaches for COVID-19 therapies.

**Table 1 cells-11-02319-t001:** Minimum Criteria for MSCs Based on The International Society for Cellular Therapy (ISCT) Recommendations.

No.	Minimum Criteria
1	Plastic adherent when maintained in standard culture conditions
2	Express specifically three specific surface antigens: CD105, CD90, and CD73
3	Lack of seven surface antigens: CD45, CD19, CD14, CD11b, CD34, CD79α, and HLA-DR
4	Can differentiate into osteoblasts, chondrocytes, and adipocytes lineage with appropriate progenitors in vitro

## Data Availability

Not applicable.

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
