# Peer review of "Potential Cell-Based and Cell-Free Therapy for Patients with COVID-19"

_cells, 2022, doi:10.3390/cells11152319_

Round 1
Reviewer 1 Report
In this review manuscript, the authors discussed the effects and mechanisms of MSC-based and cell-free therapies for patients with COVID-19. Cell-based therapies involved the direct use and injection of MSCs into the target tissue or organ. On the other hand, cell-free therapy uses the cells’ secrectomes and extracellular vesicles (EVs). Overall, this review provides useful information to understand impacts of MSC and MSC-secretomes/EVs as a therapeutic tool for COVID-19. However, there are several key issues need be addressed
1. In lines 229-267, 4.1. Secretive Therapy for COVID-19 (i.e. ARDS)
Recent studies show potential therapeutic efficacy of MSCs in respiratory diseases (i.e. COVID-19) is through paracrine actions rather than cell differentiation in (Lancet Respir Med. 2014 Dec;2(12):1016-26. doi: 10.1016/S2213-2600(14)70217-6.; Signal Transduct Target Ther. 2021 Feb 10;6(1):58. doi: 10.1038/s41392-021-00488-5.). Importantly MSC paracrine function is impaired tightly regulated by telomerase associated proteins/inflammatory pathways including critical RAP1/NFkb signalling pathway (Cell Death Discov 2015 Aug 24; 1:15007; doi: 10.1038/cddiscovery.2015.7.; Cell Death Dis. 2018 Mar 7;9(3):386.). RAP1 dysfunction impairs MSC paracrine function and immunomodulatory potency. Therefore, the quality of MSC is very important. Furthermore, MSC is also vulnerable to ischemic or hypoxic environment and undergone stem cell senesces post transplantation. Alternatively, overexpression of stress resistant factors including heme oxygenase-1 (J Cell Physiol 2019 May;234(5):7301-7319.) or ERBB4 to activate NRG1-ERBB4 signalling in MSC holds potential for preventing MSC from senesces (FASEB J. 2019 Mar;33(3):4559-4570.). it is informative to discuss such regulation mechanisms of paracrine function and vulnerability of MSC in stressful conditions, such as ARDS caused by severe COVID-19.
2. Between lines 208-218, the authors have described several problems of current adult tissue- derived MSC. It will be more informative to update progress to overcome such problems. For example, one of key challenges is the difficulty in quality control and quality assurance of MSC which was derived from different donors/tissues with variance of batch-to-batch. To avoid batch -to-batch variations including MSC quality, stem cell senescence, limited proliferative potency, recently MSCs derived from pluripotent stem cells have been proposed (i.e., Methods Mol Biol. 2016;1416:289-98.; Am J Physiol Cell Physiol. 2012 Jul 15;303(2):C115-25.;). Most recently, GMP-grade MSCs derived from hiPSCs have been used in refractory graft-versus-host-disease (GVHD) in clinical trials (Nat Med (2020). https://doi.org/10.1038/s41591-020-1050-x). PSC-MSC may provide another putative cellular source overcome many limitations of adult MSC.). It will be to informative include these contents in discussion to enhance understanding of alternative resources of MSC for covid-19.
3. EVs Therapy for COVID-19. Latest information pertaining to the stem-cells-derived exosomes and their functional role in should be discussed following latest literatures (PMID: 34901546, PMID: 34374936).
4. The conclusion is extremely small. It should be expanded further for encompassing the overall context of the manuscript. A separate section on “Future outlook” emphasizing potential research direction and persisting open questions should be discussed.
Reviewer 2 Report
This is a very comprehensive and well-written review, which may be relevant not only to the specialists in the field but also to the broader readership of the journal.
However, in the section 5. MSC-based Therapy for SARS-CoV-2 and 6. MSC-free Therapy for SARS-CoV-2 the authors should discuss in more details the mechanisms of action of either MSCs or exosomes, which can improve clinical symptoms of mild to severe form of COVID-19, e.g. their effect on suppression of cytokine storm and suppression of activated T- and NK-cells (CD3+CXCR3+ and CD56+CXCR3+). In my opinion, addition of a more detailed discussion on this issue would substantially improve the quality of the manuscript.
Round 2
Reviewer 1 Report
The revised manuscript is greatly improved . The authors have addressed the raised concerns in the updated version . I have no other questions